# Synthesis and Anti-Leishmanial Properties of Quinolones Derived from Zanthosimuline

**DOI:** 10.3390/molecules27227892

**Published:** 2022-11-15

**Authors:** Gwenaëlle Jézéquel, Laura Nogueira de Faria Cardoso, Florent Olivon, Indira Dennemont, Cécile Apel, Marc Litaudon, Fanny Roussi, Sébastien Pomel, Sandy Desrat

**Affiliations:** 1CNRS, Institut de Chimie des Substances Naturelles, Université Paris-Saclay, UPR 2301, 91198 Gif-sur-Yvette, France; 2CNRS, BioCIS, Université Paris-Saclay, 92290 Châtenay-Malabry, France

**Keywords:** quinolone, natural products, *Leishmania*

## Abstract

Quinoline derivatives and especially quinolones are considered as privileged structures in medicinal chemistry and are often associated with various biological properties. We recently isolated a series of original monoterpenyl quinolones from the bark of *Codiaeum peltatum*. As this extract was found to have a significant inhibitory activity against a *Leishmania* species, we decided to study the anti-leishmanial potential of this type of compound. Leishmaniasis is a serious health problem affecting more than 12 million people in the world. Available drugs cause harmful side effects and resistance for some of them. With the aim of finding anti-leishmanial compounds, we developed a synthetic strategy to access natural quinolones and analogues derived from zanthosimuline. We showed the versatility of this natural compound toward cyclization conditions, leading to various polycyclic quinolone-derived structures. The natural and synthetic compounds were evaluated against amastigote forms of *Leishmania infantum*. The results obtained confirmed the interest of this family of natural compounds but also revealed promising activities for some intermediates deriving from zanthosimuline. Following the same synthetic strategy, we then prepared 14 new analogues. In this work, we identified two promising molecules with good activities against intramacrophage *L. infantum* amastigotes without any cytotoxicity. We also showed that slight changes in amide functional groups affect drastically their anti-parasitic activity.

## 1. Introduction

Leishmaniasis is one of the most extended neglected tropical diseases caused by protozoan parasites from the genus *Leishmania*, counting more than twenty species and subspecies. This vector-borne disease is transmitted by the bite of an infected female phlebotomine sandfly. In its life cycle, the parasite is present in a mobile flagellated and elongated promastigote form in the sandfly, and in a non-mobile and intracellular form in the mammalian host cells [1]. Depending on the *Leishmania* species involved, the three main forms of clinical manifestations of leishmaniasis are cutaneous, mucocutaneous, and visceral, which is lethal without treatment [2]. According to the World Health Organization, between 700 000 and 1 000 000 new cases occur each year, causing the death of thousands of people. Great efforts have been made to develop new treatments, but only a few are currently available against leishmaniases. Antimoniates, paromomycin, miltefosine, and liposomal amphotericin B are broadly used, but they present several issues of toxicity, resistance, and cost [3]. To circumvent these limitations, the development of new anti-leishmanial agents is crucial.

We recently selected the bark extract of *Codiaeum peltatum* using a multi-informative molecular network approach to isolate original natural products and antiviral compounds [4]. The phytochemical investigation of this extract led to the isolation of previously undescribed compounds belonging to three different families (Figure 1): daphnane-type diterpenoid orthoesters (**1**), 1,4-dioxane-fused phenanthrene dimers (**2**), and chlorinated monoterpenyl quinolones (**3** and **4**). Among these, quinolones attracted our attention, because of their low cytotoxicity on Vero cells (CC_50_ > 200 µM) compared to the other compounds isolated [4] and because of their easy access by synthesis [5,6], facilitating the preparation of various analogs. Furthermore, quinoline derivatives are compounds of choice because they are known to possess a wide range of biological properties including anti-leishmanial activities [7,8,9,10,11,12]. They are considered as privileged structures in drug development [13,14]. In 2005, Billo et al. showed a significant inhibitory activity of the bark extract of *Codiaeum peltatum* against the promastigote form of *Leishmania donovani* with an IC_50_ value of 5 µg/mL [15]. We wondered whether this antiprotozoal activity could be due to quinolone derivatives. To address this question, we have developed the synthesis of natural chloroaustralasine A **3** and a set of analogues of the intermediate zanthosimuline **7**, then evaluated their anti-leishmanial potential.

## 2. Results and Discussion

Chloroaustralasine A **3** was prepared in three steps from commercial sources [5] by hydroxychlorination of the double bond of a tetracyclic intermediate **8** (Figure 1). This compound was obtained following a biomimetic strategy, i.e., based on an intramolecular hetero Diels–Alder cycloaddition of a geranyl quinolone. First, a Knoevenagel reaction between commercial 4-hydroxy-1-methyl-2(1H)-quinolone **5** and citral **6** allowed the introduction of a geranyl chain and spontaneously led to the oxa 6π electrocyclization product **7** [16,17]. This compound named zanthosimuline was previously isolated from the root bark of *Zanthoxylum simulans* [18]. Then, a retro oxa 6π electrocyclization/intramolecular hetero-Diels–Alder cascade gave the desired tetracyclic core **8** in a mixture of diastereomers.

To develop an efficient synthetic strategy to access chloroaustralasine A **3**, we have investigated various conditions for the retro-electrocyclization/intramolecular Diels–Alder reaction sequence (Table 1). Interestingly, this study revealed the versatility of zanthosimuline **7**. Depending on the conditions, this molecule could indeed lead to different racemic heterocycles such as the [4+2] cycloadducts **8** and **9**, the polyene cyclization compounds **10** and **11**, or the Gassman-like [2+2] cycloadduct **12**. A clear identification of all these compounds could be performed by analysis of the NMR and HRMS spectra as shown in Appendix A. The proton NMR spectrum of zanthosimuline **7** showed a triplet at 5.09 ppm and two doublet signals at 5.48 and 6.79 ppm, characteristic of the olefinic proton of the prenyl chain and the two pyranic protons, respectively. The cyclization led to the disappearance of this triplet from the prenyl chain. The subsequent formation of the double bond in compound **8** resulted in the appearance of a signal at 6.46 ppm, as well as the proton signals from the cycle junction at 3.22 and 3.55 ppm. The NMR spectra of ene cyclization products **10** and **11** were very different. Compound **10** exhibited two proton signals at 3.92 and 4.66 ppm corresponding to the protons of the hydroxypyran cycle whereas these signals disappeared for the dehydrated molecule **11**. In addition, a singlet signal corresponding to the pyran proton was observed at 6.70 ppm. Concerning the [2+2] cycloadduct **12**, two triplets at 2.46 and 2.66 ppm and a doublet at 3.12 ppm clearly appeared on the proton NMR spectrum in complete accordance with the corresponding cannabicyclol described in the literature [19,20,21].

As described in the literature [6], heating substrate **7** adsorbed on silica at 150 °C gave compound **8**, but in only 26% yield in our hands (Table 1, entry 1). Under thermic conditions, the expected Diels–Alder adduct **8** was isolated in moderate yield as a *cis*/*trans* mixture 2:1 (entry 2). Brønsted acids such as H_2_SO_4_ and PTSA (entries 3,4) mainly led to degradation. Various Lewis acids were then evaluated as promoters in this cascade reaction. Montmorillonite K10, trimethylsilyl chloride, zinc(II) chloride, and iron(III) chloride had almost no effect on zanthosimuline **7** (entries 5, 6, 10, and 11). In the presence of boron trifluoride diethyl etherate, titanium(IV) chloride, or tin(IV) chloride, the expected alkaloid **8** was not formed, but the polyene cyclization products **9** and **10** were obtained (entries 7–9). Compound **10**, that resulted from the trapping of the carbocation obtained after the cyclization [18], was obtained as a single diastereoisomer in an excellent 92% yield when using tin(IV) chloride (entry 9), while in some cases it was accompanied by the corresponding hydroxyl elimination product **11**. Original intramolecular Gassman-type cationic [2+2] cycloadduct **12** [22] was synthesized by photoredox catalysis in a diastereoselective fashion in the presence of tris(2,2-bipyridine)ruthenium(II) hexafluorophosphate under UV irradiation (entry 15), whereas aluminum(III) chloride led to rapid degradation of the substrate (entry 12). The desired isochromene analog **8** was nonetheless isolated in 55% yield together with compound **9** in 10% yield in the presence of trimethylaluminum (entry 13). This isomer **9** results from the [4+2] cycloaddition on the amide carbonyl instead of the benzophenone one. Both cyclization products **8** and **9** were obtained as a *cis*/*trans* mixture (2:1). Finally, the desired tetracyclic compound **8** was obtained quantitatively when using dimethylaluminium chloride (2 eq.) for 18 h at room temperature (entry 14) and these conditions were selected for the synthesis of chloroaustralasine A **3**. Compounds **8**, **9**, and **10** were not stable over time: **8** and **9** led to degradation and must be used directly for further functionalization while **10** dehydrated to give tetracycle **11**.

In addition to the natural quinolones **3** and **4**, we have evaluated the anti-leishmanial potential of all the stable intermediates (**7**, **11,** and **12**) prepared during the optimization of the synthetic route to **3**. These compounds were evaluated on *L. infantum* axenic and intramacrophagic amastigotes and compared to miltefosine as a reference drug (Table 2). Chloroaustralasine A **3** was the most active on axenic amastigotes of *L. infantum* with an IC_50_ value of 8 μM and showed almost no cytotoxicity on RAW 264.7 cells (Table 2, entry 1). However, no activity was observed on intramacrophage amastigotes with this compound up to a concentration of 50 µM. Although active against axenic amastigotes, the natural analog **4** was ten times more cytotoxic than **3** (entry 2). These results could partly explain the activity found for the crude ethyl acetate bark extract of *Codiaeum peltatum* on *Leishmania* parasites [15]. Zanthosimuline **7** and the pentacyclic derivative **12** revealed a moderate activity on axenic amastigotes and no activity on intramacrophagic amastigotes (entries 3 and 5). In contrast, the tetracyclic quinolone **11** showed a particularly interesting profile (entry 4) without any cytotoxicity and with a promising activity of 14.7 μM on intramacrophage amastigotes, close to the value obtained with miltefosine (entry 6). However, no activity on axenic amastigotes was observed. This compound is effective against the *L. infantum* parasite but only when the parasite is in the macrophage. This type of host-directed mechanism of action has been previously described in the literature [23,24,25,26,27].

We thus studied pharmacomodulations on the quinolone scaffold by varying the substituents on the amide function of zanthosimuline **7** and on the tetracyclic product **11**. Following the same strategy as above, the pyranoquinoline **14** was prepared, and various substituents were then introduced on its amide moiety. The resulting compounds were finally submitted to ene cyclization leading to analogs of alkaloid **11**.

The pyranoquinoline **14** [17,27] was obtained in 92% yield after condensation of 1,4-dihydroxyquinoline **13** with citral **6** catalyzed by EDDA (Figure 2). Then, various bases were evaluated to favor the *N*-alkylation toward the *O*-alkylation of the resulting amide **14**. Sodium hydride gave the lowest proportion of *O*-alkylated products (< 10%) and was chosen to introduce various alkyl (**15a–g**, **15k–l**), sulfonyl (**15h**), and acyloxy (**15i** and **15j**) substituents. All the products were obtained in good yields except the benzyl carbamate analog **15h**, which was isolated with a yield of 18% because of degradation of the substrate **14**. Whatever the base used, no acyl group could be added to the nitrogen atom. The methyl pentanoate chain of compound **15k** was hydrolyzed in basic conditions to obtain the corresponding carboxylic acid derivative **15l**. Twelve analogs **15a–l** of zanthosimuline **7** were prepared using this strategy.

To obtain the corresponding tetracyclic derivatives **16a–l**, the functionalized alkaloids **15a–l** were finally submitted to ene cyclization in the presence of tin(IV) chloride, as previously described. However, most of the compounds **15a–l** led to degradation in these conditions, and no trace of desired cyclized **16a-l** were detected, except for the benzylated derivative **16a**, which was isolated in 57% yield (Figure 3). We then tried other Lewis acids, such as boron trifluoride diethyl etherate and titanium(IV) chloride, that also enabled this type of cyclization (Table 1), but degradation was observed. It seems that this type of cyclization is substrate dependent. It gave the expected compound only with the *N*-methyl and *N*-benzyl derivatives (**7** and **15a**).

All the compounds prepared (**14**, **15a–l**, and **16a**) were evaluated on *L. infantum* axenic and intramacrophage amastigotes and compared to miltefosine as a reference drug (Table 3).

All compounds but **15g** and **15h** (entries 8 and 9) displayed better activities than compound **7** against axenic amastigotes with IC_50_ between 7 and 22 µM. However, most of them do not act on the intramacrophage parasites and have a low selectivity index. In this series, only zanthosimuline analogs **15a** and **15k** show a different profile and inhibit the intramacrophage parasite at 23.7 µM and 19.0 µM, respectively (entries 2 and 12). The benzylated derivative **15a** is the most promising compound with a good selectivity index, above 4.2 (entry 2). In comparison with zanthosimuline **7**, the addition of a benzyl group is beneficial for the anti-leishmanial activity. However, contrary to compound **11**, the cyclization of the alkenyl chain of the benzylated derivative **15a** giving compound **16a** led to the loss of activity on the intramacrophage form of the parasite and therefore to a low selectivity index (entry 14).

This study reveals the potential of zanthosimuline-derived quinolones against *Leishmania* parasites. On one hand, the natural chloroaustralasine compounds (**3** and **4**) could, at least partly, be responsible for the activity previously observed with the crude ethyl acetate bark extract of *Codiaeum peltatum*. On the other hand, some by-products obtained in developing a synthetic route toward the natural quinolones showed interesting therapeutic profiles. For example, alkaloid **11** showed an interesting inhibitory activity of 14.7 μM on intramacrophage *L. infantum* amastigotes without any cytotoxicity. With a selectivity index over 6.8, this compound (**11**) displayed anti-leishmanial properties on intramacrophage amastigotes close to miltefosine, the reference compound. We have also shown that small modifications of the amide function can have an impact on their anti-parasitic activity. Whereas the addition of a benzyl group on the zanthosimuline core improved the growth inhibition of *L. infantum*, other functionalization led to the loss of any activity.

## 3. Materials and Methods

### 3.1. General Experimental Details

All reagents and solvents were used as purchased from commercial suppliers or were purified/dried according to Armarego and Chai [28]. Purifications by column chromatography on silica gel were performed using Merck Silica Gel 60 (70–230 mesh). ^1^H and ^13^C NMR spectra were recorded on Bruker ARX500 instruments using CDCl_3_ as an internal reference. Chemical shifts (*δ* values) are given in parts per million (ppm), and the multiplicity of signals is reported as follows: s, singlet; bs, broad singlet; d, doublet; t, triplet; q, quartet; dd, doublet of doublets; m, multiplet. HRMS analyses were performed using a Waters LCT Premier instrument by positive ion mode ElectroSpray Ionization (ESI+). A Monowave 50 appartus from Anton-Paar was used with a thermic control (Pmax: 315 W).

### 3.2. Procedures and Analytical Description of Compounds

Zanthosimuline **7**. A suspension of 4-hydroxy-1-methyl-2-(1*H*)-quinolone **5** (1.50 g, 8.56 mmol, 1 eq.), citral **6** (1.47 mL, 8.56 mmol, 1 eq.), and EDDA (0.30 g, 1.71 mmol, 0.2 eq.) in anhydrous EtOH (15 mL) was heated under MW irradiation at 150 °C for 20 min. The mixture was concentrated under reduced pressure and the desired compound purified by flash chromatography on silica gel (80 g) using heptane/EtOAc from 10:0 to 5:5 to give zanthosimuline **7** as a light-yellow oil (2.45 g, 7.91 mmol, 92%). ^1^H NMR (500 MHz, CDCl_3_, 25 °C): *δ* 7.93 (d, *J* = 7.9 Hz, 1H), 7.52 (t, *J* = 8.0 Hz, 1H), 7.29 (d, *J* = 8.0 Hz, 1H), 7.20 (t, *J* = 8.0 Hz, 1H), 6.77 (d, *J* = 10.0 Hz, 1H), 5.46 (d, *J* = 10.0 Hz, 1H), 5.07 (t, *J* = 7.0 Hz, 1H), 3.67 (s, 3H), 2.12 (q, *J* = 7.0 Hz, 2H), 1.82–1.62 (m, 2H), 1.60 (s, 3H), 1.52 (s, 3H), 1.45 (s, 3H) ppm. ^13^C NMR (125 MHz, CDCl_3_, 25 °C): *δ* 161.2, 155.6, 139.5, 132.2, 131.0, 125.4, 123.9, 123.3, 121.9, 118,6, 116.2, 114.2, 105.7, 81.5, 41.7, 29.5, 27.2, 25.8, 22.8, 17.8 ppm. HRMS (ESI+): calcd. for C_20_H_24_NO_2_^+^ [M+H]^+^ 310.1802, found 310.1804. Analytical data were consistent with those described by Wu and Chen [18].

6,6,9,12-Tetramethyl-6,6a,7,8,10a,12-hexahydro-11*H*-isochromeno [4,3-*c*]quinolin-11-one (**8**). To a solution of zanthosimuline **7** (198 mg, 0.64 mmol, 1 eq.) in anhydrous dichloromethane (6 mL) under Ar atm. was added a 1 M solution of Me_2_AlCl in hexanes (1.28 mL, 1.28 mmol, 2 eq.). The reaction mixture was stirred at RT for 18 h before adding 0.5 mL of water. The mixture was dried over MgSO_4_, filtrated over a Celite^®^ pad, and concentrated under reduced pressure to give compound **8** in a *cis*/*trans* mixture (2:1) as a pale-yellow oil (196 mg, 0.64 mmol, 95%). ^1^H NMR of ***cis*-8** (500 MHz, CDCl_3_, 25 °C): *δ* 7.93 (dd, *J* = 8.0, 1.3 Hz, 1H), 7.48 (bt, *J* = 8.0 Hz, 1H), 7.27 (d, *J* = 8.4 Hz, 1H), 7.17 (bt, *J* = 7.7 Hz, 1H), 6.47−6.41 (m, 1H), 3.66 (s, 3H), 3.55 (bs, 1H), 2.03−1.87 (m, 3H), 1.87−1.75 (m, 1H), 1.69 (bs, 3H), 1.51 (s, 3H), 1.44−1.32 (m, 1H), 1.31 (s, 3H) ppm. ^1^H NMR of *trans*-**8** (500 MHz, CDCl_3_, 25 °C): *δ* 7.93 (dd, *J* = 8.0, 1.3 Hz, 1H), 7.48 (bt, *J* = 8.0 Hz, 1H), 7.27 (d, *J* = 8.4 Hz, 1H), 7.17 (bt, *J* = 7.7 Hz, 1H), 6.47−6.41 (m, 1H), 3.67 (s, 3H), 3.24–3.19 (m, 1H), 2.23−2.12 (m, 2H), 2.03−1.87 (m, 1H), 1.75−1.62 (m, 1H), 1.69 (bs, 3H), 1.53 (s, 3H), 1.54−1.43 (m, 1H), 1.14 (s, 3H) ppm. HRMS (ESI+): calcd. for C_20_H_24_NO_2_^+^ [M+H]^+^ 310.1802, found 310.1812. Analytical data were consistent with those described by Riveira et al. [6].

Chloroaustralasine A **3**. To a solution of *cis*/*trans*-**8** (44 mg, 0.14 mmol, 1 eq.) in a 1:3 mixture of *tert*-butanol/citric acid buffer (0.1 M, pH 3.5, 14 mL) were added an aqueous suspension of chloroperoxidase (3000 U/mL) from *Caldariomyces fumago* (40 *µ*L, 120 U) and NaCl (8 mg, 0.14 mmol, 1 eq.). Then, a 30% H_2_O_2_ aqueous solution (20 µL, 0.16 mmol, 1 eq.) was added portion wise every 10 min for 1 h. The products were then extracted twice with MTBE (14 mL). The combined organic phases were dried over MgSO_4_ and concentrated in vacuo. The residue was purified by semi-preparative HPLC (Kinetex C_18_, H_2_O-MeCN 45:55 at 4.7 mL·min^−1^) to afford chloroaustralasine A **3** as a white powder (12 mg, 0.033 mmol, 24%, *t*_R_: 13.3 min). ^1^H NMR (500 MHz, CDCl_3_, 25 °C): *δ* 7.98 (dd, *J* = 8.0, 1.4 Hz, 1H), 7.55 (ddd, *J* = 8.5, 7.1, 1.4 Hz, 1H), 7.34 (d, *J* = 8.5 Hz, 1H), 7.23 (dd, *J* = 8.0, 7.1 Hz, 1H), 5.81 (s, 1H), 3.70 (s, 3H), 3.60 (dd, *J* = 11.4, 2.6 Hz, 1H), 2.17 (ddd, *J* = 12.0, 11.4, 3.6 Hz 1H), 1.93 (ddd, *J* = 13.7, 12.9, 5.1 Hz, 1H), 1.72–1.46 (m, 3H), 1.53 (s, 3H), 1.43 (s, 3H), 1.18 (s, 3H) ppm. ^13^C NMR (125 MHz, CDCl_3_, 25 °C): *δ* 162.9, 157.5, 139.0, 130.6, 123.4, 121.8, 116.9, 114.1, 106.3, 79.8, 72.4, 65.5, 40.1, 36.0, 32.9, 29.6, 29.4, 27.6, 22.9, 19.8 ppm. HRMS (ESI+): calcd. for C_20_H_25_ClNO_3_^+^ [M+H]^+^ 362.1515, found 362.1515. [5]

7-Hydroxy-5,8,8,11a-tetramethyl-5,7,7a,8,9,10,11,11a-octahydro-6*H*-chromeno [3,2-*c*]quinolin-6-one (**10**). To a solution of zanthosimuline **7** (350 mg, 1.13 mmol, 1 eq.) in anhydrous dichloromethane (10 mL) under Ar atm. was added dropwise BF_3_.OEt_2_ (280 µL, 2.26 mmol, 2 eq.). After 18 h at RT, the mixture was quenched with water (0.3 mL). MgSO_4_ was added to the solution and the solids were eliminated by filtration over a Celite^®^ pad. The filtrate was concentrated under reduced pressure and purified by flash chromatography on silica gel (24 g) with heptane/EtOAc from 10:0 to 0:10 to give compound **11** as a white solid (98 mg, 0.32 mmol, 28%) and compound **10** as a yellow gum (112 mg, 0.334 mmol, 30%). ^1^H NMR (500 MHz, CDCl_3_, 25 °C): *δ* 8.02 (d, *J* = 8.0 Hz, 1H), 7.58 (t, *J* = 7.5 Hz, 1H), 7.37 (d, *J* = 8.0 Hz, 1H), 7.26 (t, *J* = 7.5 Hz, 1H), 4.78 (s, 1H), 3.73 (s, 3H), 2.24 (d, *J* = 14.5 Hz, 1H), 1.83 (qt, *J* = 13.8, 3.0 Hz, 1H), 1.74 (s, 1H), 1.62 (td, *J* = 13.8, 4.0 Hz, 1H), 1.55–1.42 (m, 2H), 1.42 (s, 3H), 1.33 (td, *J* = 13.8, 3.0 Hz, 1H), 1.29–1.23 (m, 1H), 1.11 (s, 3H), 0.60 (s, 3H) ppm. ^13^C NMR (125 MHz, CD_3_CN, 25 °C): *δ* 164.3, 156.4, 140.1, 132.0, 124.0, 122.7, 117.0, 115.5, 110.0, 79.1, 62.9, 52.6, 41.7, 39.6, 33.5, 32.0, 29.3, 26.9, 21.3, 18.6 ppm. HRMS (ESI+): calcd. for C_20_H_25_NO_3_^+^ [M+H]^+^ 328.1907, found 328.1897. Compound **10** was found to be unstable over a period of one week, giving compound **11**.

5,8,8,11a-Tetramethyl-5,8,9,10,11,11a-hexahydro-6*H*-chromeno [3,2-*c*]quinolin-6-one (**11**). To a solution of zanthosimuline **7** (50 mg, 0.16 mmol, 1 eq.) in anhydrous dichloromethane (5 mL) under Ar atm. was added SnCl_4_ (84 mg, 0.32 mmol, 2 eq.). After 1 h at RT, the mixture was quenched with water (0.2 mL). MgSO_4_ was added to the solution and the solids were eliminated by filtration over a Celite^®^ pad. The filtrate was concentrated under reduced pressure and purified by flash chromatography on silica gel (24 g) with heptane/EtOAc from 10:0 to 5:5 to give compound **11** as a white solid (46 mg, 0.15 mmol, 95%). ^1^H NMR (500 MHz, CDCl_3_, 25 °C): *δ* 7.91 (d, *J* = 7.9 Hz, 1H), 7.50 (t, *J* = 8.0 Hz, 1H), 7.30 (d, *J* = 8.0 Hz, 1H), 7.20 (t, *J* = 8.0 Hz, 1H), 6.68 (d, *J* = 10.0 Hz, 1H), 3.69 (s, 3H), 2.19 (m, 1H), 1.97 (dt, *J* = 12.9, 5.0 Hz, 1H), 1.72–1.69 (m, 2H), 1.50−1.43 (m, 1H), 1.44 (s, 3H), 1.28 (s, 3H), 1.17 (s, 3H) ppm. ^13^C NMR (125 MHz, CDCl_3_, 25 °C): *δ* 161.4, 153.9, 144.4, 139.2, 130.6, 123.1, 121.9, 116.4, 114.2, 112.4, 108.1, 80.8, 40.1, 39.6, 36.1, 31.0, 30.8, 29.5, 25.8, 19.2 ppm. HRMS (ESI+): calcd. for C_20_H_24_NO_2_^+^ [M+H]^+^ 310.1802, found 310.1799.

1,1,3a,9-Tetramethyl-1,1a,1a1,2,3,3a,9,10b-octahydro-10*H*-4-oxa-9-azacyclobuta [7,1]indeno [5,6-*a*]naphthalen-10-one (**12**). To a solution of zanthosimuline **7** (50 mg, 0.16 mmol, 1 eq.) in anhydrous nitromethane (1 mL) under Ar atm. were added Ru(bpy)_3_(PF_6_)_2_ (7 mg, 0.01 mmol, 5 mol%) and methyl viologen (6 mg, 0.02 mmol, 10 mol%). After 1 h under blue light irradiation at RT, the mixture was concentrated under reduced pressure and the product was purified by flash chromatography on silica gel (24 g) with heptane/EtOAc from 10:0 to 5:5 to give compound **12** as a yellow gum (43 mg, 0.14 mmol, 86%). ^1^H NMR (500 MHz, CDCl_3_, 25 °C): *δ* 8.00 (dd, *J* = 8.0, 1.4 Hz, 1H), 7.53 (td, *J* = 8.0, 1.4 Hz, 1H), 7.32 (d, *J* = 8.0 Hz, 1H), 7.21 (t, *J* = 8.0 Hz, 1H), 3.68 (s, 3H), 3.13 (d, *J* = 9.2 Hz, 1H), 2.66 (dd, *J* = 9.8, 7.6 Hz, 1H), 2.46 (t, *J* = 7.6 Hz, 1H), 2.04–1.95 (m 1H), 1.79–1.69 (m, 2H), 1.67–1.58 (m, 1H), 1.51 (s, 3H), 1.46 (s, 3H), 0.84 (s, 3H) ppm. ^13^C NMR (125 MHz, CDCl_3_, 25 °C): *δ* 163.5, 155.1, 138.8, 130.0, 122.9, 121.3, 117.0, 113.7, 109.1, 85.2, 46.8, 39.5, 39.1, 38.0, 36.9, 33.8, 29.2, 27.5, 26.0, 17.6 ppm. HRMS (ESI+): calcd. for C_20_H_24_NO_2_^+^ [M+H]^+^ 310.1802, found 310.1807 [21].

2-Methyl-2-(4-methylpent-3-en-1-yl)-2,6-dihydro-5*H*-pyrano [3,2-*c*]quinolin-5-one (**14**). A suspension of 1,4-dihydroxyquinoline **13** (3.00 g, 18.6 mmol, 1 eq.), citral **6** (3.19 mL, 18.6 mmol, 1 eq.), and EDDA (0.66 g, 3.72 mmol, 0.2 eq.) in anhydrous EtOH (40 mL) was heated under MW irradiation at 150 °C for 30 min. After decanting for 10 min, the supernatant containing the expected compound was removed. The solids were rinsed twice with EtOH. The combined supernatants were then concentrated under reduced pressure to obtain a light-orange solid (4.78 g, 16.2 mmol, 87%). ^1^H NMR (500 MHz, CDCl_3_, 25 °C): *δ* 11.57 (bs, 1H), 7.87 (dd, *J* = 8.1, 1.4 Hz, 1H), 7.47 (td, *J* = 7.2, 1.2 Hz, 1H), 7.34 (d, *J* = 7.9 Hz, 1H), 7.19 (t, *J* = 7.9, 1H), 6.81 (d, *J* = 10.1 Hz, 1H), 5.50 (d, *J* = 10.1 Hz, 1H), 5.10 (tt, *J* = 7.1, 1.3 Hz, 1H), 2.16 (q, *J* = 7.9 Hz, 2H), 1.89–1.73 (m, 2H), 1.63 (s, 3H), 1.55 (s, 3H), 1.51 (s, 3H) ppm. ^13^C NMR (125 MHz, CDCl_3_, 25 °C): δ 163.1, 157.6, 138.3, 132.0, 130.8, 125.1, 123.9, 122.5, 122.1, 117.8, 116.3, 115.2, 105.4, 81.7, 41.7, 27.2, 25.7, 22.7, 17.7 ppm. HRMS (ESI+): calcd. for C_19_H_21_NO_2_^+^ [M+H]^+^ 296.1645, found 296.1646. Analytical data were consistent with those described in the literature [17,27].

General procedure for functionalization of compound **14**. To a solution of compound **14** (50 mg, 0.17 mmol, 1 eq.) in anhydrous DMF (2.5 mL), at 0 °C and under Ar atm., was added NaH (60% in mineral oil, 8 mg, 0.20 mmol, 1.2 eq.). The suspension was stirred for 30 min at 0 °C and the electrophile was added (1–2 eq.). The reaction mixture was allowed to warm to RT and stirred for 6 to 18 h. A saturated solution of NH_4_Cl was then added and the product was extracted with EtOAc (3 times). The combined organic phases were dried over MgSO_4_, filtered, and concentrated under reduced pressure. A purification by flash column chromatography on silica gel with heptane/EtOAc as eluent gave the desired pure compound.

6-Benzyl-2-methyl-2-(4-methylpent-3-en-1-yl)-2,6-dihydro-5*H*-pyrano [3,2-*c*]quinolin-5-one (**15a**). Following the general procedure described with benzyl bromide as the electrophile, compound **15a** was isolated as a light-yellow oil (43 mg, 0.11 mmol, 66%). ^1^H NMR (500 MHz, CDCl_3_, 25 °C): *δ* 7.96 (dd, *J* = 8.0, 1.3 Hz, 1H), 7.37 (dt, *J* = 8.0, 1.3 Hz, 1H), 7.27 (d, *J* = 8.0 Hz, 2H), 7.23–7.18 (m, 4H), 7.15 (t, *J* = 7.3 Hz, 1H), 6.84 (d, *J* = 10.0 Hz, 1H), 5.50 (d, *J* = 10.0 Hz, 1H), 5.11 (t, *J* = 7.0 Hz, 1H), 2.16 (q, *J* = 7.0 Hz, 2H), 1.91–1.84 (m, 1H), 1.80–1.73 (m, 1 H), 1.62 (s, 3H), 1.56 (s, 3H), 1.50 (s, 3H) ppm. ^13^C NMR (125 MHz, CDCl_3_, 25 °C): *δ* 161.1, 155.7, 138.8, 136.8, 131.9, 130.8, 128.7 (2C), 127.1, 126.5 (2C), 125.2, 123.7, 123.1, 121.7, 118.4, 116.2, 114.8, 105.2, 81.5, 45.7, 41.6, 27.1, 25.6, 22.6, 17.6 ppm. HRMS (ESI+): calcd. for C_26_H_27_NO_2_^+^ [M+H]^+^ 386.2115, found 386.2130.

6-(3-Methoxybenzyl)-2-methyl-2-(4-methylpent-3-en-1-yl)-2,6-dihydro-5*H*-pyrano [3,2-*c*]quinolin-5-one (**15b**). Following the general procedure described with 3-methoxybenzyl bromide as the electrophile, compound **15b** was isolated as a light-yellow oil (45 mg, 0.11 mmol, 67%). ^1^H NMR (500 MHz, CDCl_3_, 25 °C): 7.96 (dd, *J* = 8.0, 1.3 Hz, 1H), 7.37 (dt, *J* = 8.0, 1.3 Hz, 1H), 7.23–7.15 (m, 3H), 6.84 (d, *J* = 10.0 Hz, 1H), 6.80 (d, *J* = 7.8 Hz, 1H), 6.78–6.74 (m, 2H), 5.51 (d, *J* = 10.0 Hz, 1H), 5.12 (t, *J* = 7.0 Hz, 1H), 3.75 (s, 3H), 2.17 (q, *J* = 7.0 Hz, 2H), 1.92–1.85 (m, 1H), 1.81–1.74 (m, 1 H), 1.64 (s, 3H), 1.57 (s, 3H), 1.51 (s, 3H) ppm. ^13^C NMR (125 MHz, CDCl_3_, 25 °C): *δ* 161.1, 160.0, 155.7, 138.9, 138.5, 132.0, 130.8, 129.8, 125.2, 123.7, 123.1, 121.8, 118.8, 118.4, 116.2, 114.9, 112.6, 112.2, 105.2, 81.5, 55.2, 45.7, 41.6, 27.1, 25.6, 22.6, 17.6 ppm. HRMS (ESI+): calcd. for C_27_H_30_NO_3_^+^ [M+H]^+^ 416.2220, found 416.2229.

2-Methyl-2-(4-methylpent-3-en-1-yl)-6-(pyridin-2-ylmethyl)-2,6-dihydro-5*H*-pyrano [3,2-*c*]quinolin-5-one (**15c**). Following the general procedure described with 2-(bromomethyl)pyridine as the electrophile, compound **15c** was isolated as a yellow oil (53 mg, 0.14 mmol, 80%). ^1^H NMR (500 MHz, CDCl_3_, 25 °C): *δ* 8.57 (d, *J* = 4.8 Hz, 1H), 7.95 (dd, *J* = 8.0, 1.3 Hz, 1H), 7.55 (dt, *J* = 8.0, 1.3 Hz, 1H), 7.43–7.35 (m, 2H), 7.20–7.12 (m, 3H), 6.84 (d, *J* = 10.0 Hz, 1H), 5.65 (bs, 2H), 5.52 (d, *J* = 10.0 Hz, 1H), 5.11 (t, *J* = 7.0 Hz, 1H), 2.16 (q, *J* = 7.0 Hz, 2H), 1.91–1.84 (m, 1H), 1.80–1.73 (m, 1 H), 1.63 (s, 3H), 1.56 (s, 3H), 1.50 (s, 3H) ppm. ^13^C NMR (125 MHz, CDCl_3_, 25 °C): *δ* 161.1, 157.1, 155.9, 149.2, 138.9, 136.9, 132.0, 130.9, 125.2, 123.7, 123.1, 122.3, 121.9, 121.5, 118.3, 116.2, 115.1, 105.2, 81.6, 48.0, 41.6, 27.1, 25.6, 22.6, 17.6 ppm. HRMS (ESI+): calcd. for C_25_H_27_N_2_O_2_^+^ [M+H]^+^ 387.2067, found 387.2064.

6-(2-Methoxyethyl)-2-methyl-2-(4-methylpent-3-en-1-yl)-2,6-dihydro-5*H*-pyrano [3,2-*c*]quinolin-5-one (**15d**). Following the general procedure described with 2-bromomethyl methyl ether as the electrophile, compound **15d** was isolated as a yellow oil (37 mg, 0.11 mmol, 64%). ^1^H NMR (500 MHz, CDCl_3_, 25 °C): *δ* 7.95 (dd, *J* = 8.0, 1.2 Hz, 1H), 7.54–7.50 (m, 1H), 7.49 (t, *J* = 8.0 Hz, 1H), 7.23 (td, *J* = 8.0, 1.0 Hz, 1H), 6.80 (d, *J* = 10.0 Hz, 1H), 5.48 (d, *J* = 10.0 Hz, 1H), 5.09 (t, *J* = 6.9 Hz, 1H), 4.47 (t, *J* = 6.1 Hz, 2H), 3.71 (t, *J* = 6.1 Hz, 2H), 3.36 (s, 3H), 2.14 (q, *J* = 7.8 Hz, 2H), 1.89–1.81 (m, 1H), 1.78–1.68 (m, 1H), 1.62 (s, 3H), 1.55 (s, 3H), 1.48 (s, 3H) ppm. ^13^C NMR (125 MHz, CDCl_3_, 25 °C): *δ* 160.9, 155.6, 139.2, 132.0, 130.7, 125.2, 123.8, 123.1, 121.7, 118.2, 116.1, 114.6, 105.2, 81.4, 70.1, 59.1, 42.1, 41.6, 27.1, 25.6, 22.6, 17.6 ppm. HRMS (ESI+): calcd. for C_22_H_28_NO_3_^+^ [M+H]^+^ 354.2064, found 354.2060.

2-Methyl-6-(3-methylbut-2-en-1-yl)-2-(4-methylpent-3-en-1-yl)-2,6-dihydro-5*H*-pyrano [3,2-*c*]quinolin-5-one (**15e**). Following the general procedure described with prenyl bromide as the electrophile, compound **15e** was isolated as a yellow oil (45 mg, 0.13 mmol, 75%). ^1^H NMR (500 MHz, CDCl_3_, 25 °C): *δ* 7.95 (dd, *J* = 8.0, 1.2 Hz, 1H), 7.50 (t, *J* = 7.6 Hz, 1H), 7.49 (t, *J* = 8.0 Hz, 1H), 7.26 (d, *J* = 8.0 Hz, 1H), 7.20 (d, *J* = 8.0 Hz, 1H), 6.80 (d, *J* = 10.0 Hz, 1H), 5.45 (d, *J* = 10.0 Hz, 1H), 5.14 (t, *J* = 6.0 Hz, 1H), 5.09 (t, *J* = 6.9 Hz, 1H), 4.91 (d, *J* = 6.0 Hz, 2H), 2.14 (q, *J* = 7.7 Hz, 2H), 1.89 (s, 3H), 1.89–1.81 (m, 1H), 1.78–1.68 (m, 1H), 1.71 (s, 3H), 1.62 (s, 3H), 1.55 (s, 3H), 1.47 (s, 3H) ppm. ^13^C NMR (125 MHz, CDCl_3_, 25 °C): *δ* 160.7, 155.4, 138.8, 135.6, 131.9, 130.6, 125.2, 123.8, 123.1, 121.5, 119.8, 118.4, 116.2, 114.5, 104.4, 81.2, 41.6, 40.6, 27.0, 25.6 (2C), 22.7, 18.3, 17.6 ppm. HRMS (ESI+): calcd. for C_24_H_30_NO_2_^+^ [M+H]^+^ 364.2271, found 364.2276.

2-Methyl-2-(4-methylpent-3-en-1-yl)-6-phenethyl-2,6-dihydro-5*H*-pyrano [3,2-*c*]quinolin-5-one (**15f**). Following the general procedure described with 4-phenethyl bromide as the electrophile, compound **15f** was isolated as a yellow oil (42 mg, 0.11 mmol, 62%). ^1^H NMR (500 MHz, CDCl_3_, 25 °C): *δ* 7.99 (dd, *J* = 7.5, 1.3 Hz, 1H), 7.55 (t, *J* = 7.5 Hz, 1H), 7.39–7.31 (m, 6H), 7.22 (t, *J* = 8.0 Hz, 1H), 6.81 (d, *J* = 10.0 Hz, 1H), 5.50 (d, *J* = 10.0 Hz, 1H), 5.11 (t, *J* = 6.0 Hz, 1H), 4.49–4.44 (m, 2H), 3.04–2.99 (m, 2H), 2.16 (q, *J* = 7.7 Hz, 2H), 1.90–1.82 (m, 1H), 1.80–1.72 (m, 1H), 1.63 (s, 3H), 1.56 (s, 3H), 1.50 (s, 3H) ppm. ^13^C NMR (125 MHz, CDCl_3_, 25 °C): *δ* 160.6, 155.4, 138.6, 138.5, 132.0, 130.8, 128.8 (2C), 128.7 (2C), 126.6, 125.4, 123.8, 123.4, 121.6, 118.2, 116.2, 113.8, 105.4, 81.4, 43.8, 41.6, 34.0, 27.1, 25.6, 22.6, 17.6 ppm. HRMS (ESI+): calcd. for C_27_H_30_NO_2_^+^ [M+H]^+^ 400.2271, found 400.2266.

2-Methyl-2-(4-methylpent-3-en-1-yl)-6-pentyl-2,6-dihydro-5*H*-pyrano [3,2-*c*]quinolin-5-one (**15g**). Following the general procedure described with 1-bromopentane as the electrophile, compound **15g** was isolated as a yellow oil (51 mg, 0.14 mmol, 82%). ^1^H NMR (500 MHz, CDCl_3_, 25 °C): *δ* 7.96 (dd, *J* = 8.0, 1.2 Hz, 1H), 7.52 (t, *J* = 7.6 Hz, 1H), 7.29 (d, *J* = 8.0 Hz, 1H), 7.20 (d, *J* = 8.0 Hz, 1H), 6.80 (d, *J* = 10.0 Hz, 1H), 5.47 (d, *J* = 10.0 Hz, 1H), 5.10 (t, *J* = 6.0 Hz, 1H), 4.25 (dd, *J* = 10.5, 7.6 Hz, 2H), 2.14 (q, *J* = 7.7 Hz, 2H), 1.89–1.81 (m, 1H), 1.78–1.68 (m, 3H), 1.62 (s, 3H), 1.55 (s, 3H), 1.47 (s, 3H), 1.46–1.36 (m, 4H), 0.93 (t, *J* = 7.6 Hz, 3H) ppm. ^13^C NMR (125 MHz, CDCl_3_, 25 °C): *δ* 160.7, 155.2, 138.4, 131.9, 130.7, 125.2, 123.8, 123.2, 121.4, 118.4, 116.2, 114.0, 105.4, 81.2, 42.2, 41.6, 29.2, 27.4, 27.0, 25.6, 22.6, 22.5, 17.6, 14.0 ppm. HRMS (ESI+): calcd. for C_24_H_32_NO_2_^+^ [M+H]^+^ 366.2428, found 366.2428.

2-Methyl-2-(4-methylpent-3-en-1-yl)-6-(phenylsulfonyl)-2,6-dihydro-5*H*-pyrano [3,2-*c*]quinolin-5-one (**15h**). Following the general procedure described with 1-bromopentane as the electrophile, compound **15h** was isolated as a yellow oil (34 mg, 0.08 mmol, 47%). ^1^H NMR (500 MHz, CDCl_3_, 25 °C): *δ* 8.18 (d, *J* = 7.5Hz, 2H), 8.0 (dd, *J* = 7.5, 1.3 Hz, 1H), 7.69–7.63 (m, 2H), 7.60–7.55 (m, 3H), 7.41 (t, *J* = 8.0 Hz, 1H), 6.67 (d, *J* = 10.0 Hz, 1H), 5.51 (d, *J* = 10.0 Hz, 1H), 5.07 (t, *J* = 6.0 Hz, 1H), 2.14 (q, *J* = 7.7 Hz, 2H), 1.91–1.83 (m, 1H), 1.81–1.72 (m, 1H), 1.60 (s, 3H), 1.51 (s, 3H), 1.25 (s, 3H) ppm. ^13^C NMR (125 MHz, CDCl_3_, 25 °C): *δ* 158.7, 152.2, 145.7, 137.6, 132.2, 130.3, 129.1 (2C), 128.8 (2C), 128.2, 128.1, 125.6, 123.5, 121.8, 118.8, 116.5, 103.7, 82.0, 41.7, 29.7, 27.3, 25.6, 22.5, 17.6 ppm. HRMS (ESI): calcd. for C_25_H_26_NO_4_S^+^ [M+H]^+^ 436.1577, found 436.1570.

Phenyl 2-methyl-2-(4-methylpent-3-en-1-yl)-5-oxo-2*H*-pyrano [3,2-*c*]quinoline-6(5*H*)-carboxylate (**15i**). Following the general procedure described with phenyl chloroformate as the electrophile, compound **15i** was isolated as a yellow oil (13 mg, 0.03 mmol, 18%). ^1^H NMR (500 MHz, CDCl_3_, 25 °C): *δ* 7.97 (dd, *J* = 8.0, 1.3 Hz, 1H), 7.55 (dt, *J* = 8.0, 1.3 Hz, 1H), 7.50–7.45 (m, 2H), 7.44–7.40 (m, 2H), 7.36–7.25 (m, 3H), 6.74 (d, *J* = 10.0 Hz, 1H), 5.52 (d, *J* = 10.0 Hz, 1H), 5.10 (t, *J* = 7.0 Hz, 1H), 2.16 (q, *J* = 7.0 Hz, 2H), 1.93–1.86 (m, 1H), 1.81–1.74 (m, 1 H), 1.63 (s, 3H), 1.57 (s, 3H), 1.52 (s, 3H) ppm. ^13^C NMR (125 MHz, CDCl_3_, 25 °C): *δ* 159.5, 157.2, 152.1, 151.0, 135.8, 132.2, 131.3, 129.8 (2C), 126.9, 125.5, 123.5, 123.4, 123.3, 120.9 (2C), 117.1, 115.5, 113.9, 104.9, 82.4, 41.7, 27.3, 25.6, 22.6, 17.6 ppm. HRMS (ESI+): calcd. for C_26_H_26_NO_4_^+^ [M+H]^+^ 416.1856, found 416.1854.

*tert*-Butyl 2-methyl-2-(4-methylpent-3-en-1-yl)-5-oxo-2*H*-pyrano [3,2-*c*]quinoline-6(5*H*)-carboxylate (**15j**). Following the general procedure described with phenyl chloroformate as the electrophile, compound **15j** was isolated as a yellow oil (49 mg, 0.13 mmol, 74%). ^1^H NMR (500 MHz, CDCl_3_, 25 °C): *δ* 8.07 (d, *J* = 8.2 Hz, 1H), 7.86 (d, *J* = 8.2 Hz, 1H), 7.62 (dt, *J* = 7.4, 1.3 Hz, 1H), 7.43 (t, *J* = 7.4 Hz, 1H), 6.49 (d, *J* = 10.0 Hz, 1H), 5.61 (d, *J* = 10.0 Hz, 1H), 5.09 (t, *J* = 7.0 Hz, 1H), 2.16 (q, *J* = 7.2 Hz, 2H), 1.92–1.85 (m, 1H), 1.82–1.75 (m, 1 H), 1.61 (s, 3H), 1.57 (s, 9H), 1.53 (s, 3H), 1.52 (s, 3H) ppm. ^13^C NMR (125 MHz, CDCl_3_, 25 °C): *δ* 158.5, 153.0, 150.6, 146.3, 132.1, 130.2, 128.4, 128.1, 125.4, 123.5, 121.7, 119.9, 116.5, 104.3, 84.0, 81.6, 41.7, 27.7 (3C), 27.2, 25.5, 22.5, 17.5 ppm. HRMS (ESI+): calcd. for C_24_H_30_NO_4_^+^ [M+H]^+^ 396.2169, found 396.2177.

Methyl 5-(2-methyl-2-(4-methylpent-3-en-1-yl)-5-oxo-2*H*-pyrano [3,2-*c*]quinolin-6(5*H*)-yl)pentanoate (**15k**). Following the general procedure described with methyl 5-bromovalerate as the electrophile, compound **15k** was isolated as a yellow oil (41 mg, 0.10 mmol, 59%). ^1^H NMR (500 MHz, CDCl_3_, 25 °C): *δ* 7.96 (dd, *J* = 8.0, 1.2 Hz, 1H), 7.53 (t, *J* = 7.6 Hz, 1H), 7.29 (d, *J* = 8.0 Hz, 1H), 7.20 (t, *J* = 8.0 Hz, 1H), 6.77 (d, *J* = 10.0 Hz, 1H), 5.48 (d, *J* = 10.0 Hz, 1H), 5.10 (t, *J* = 6.0 Hz, 1H), 4.28 (t, *J* = 6.6 Hz, 2H), 3.66 (s, 3H), 2.40 (t, *J* = 6.6 Hz, 2H), 2.14 (q, *J* = 7.7 Hz, 2H), 1.88–1.70 (m, 6H), 1.62 (s, 3H), 1.55 (s, 3H), 1.47 (s, 3H) ppm. ^13^C NMR (125 MHz, CDCl_3_, 25 °C): *δ* 173.8, 160.7, 155.3, 138.5, 131.9, 130.8, 125.3, 123.8, 123.3, 121.6, 118.3, 116.2, 113.9, 105.3, 81.3, 51.3, 41.7, 41.6, 33.7, 27.2, 27.1, 25.6, 22.6, 22.4, 17.6 ppm. HRMS (ESI+): calcd. for C_25_H_32_NO_4_^+^ [M+H]^+^ 410.2326, found 410.2322.

5-(2-Methyl-2-(4-methylpent-3-en-1-yl)-5-oxo-2*H*-pyrano [3,2-*c*]quinolin-6(5*H*)-yl)pentanoic acid (**15l**). To a solution of ester derivative **15k** (25 mg, 0.06 mmol, 1 eq.) in a 1:1 mixture of MeOH/H_2_O was added LiOH.H_2_O (7 mg, 0.18 mmol, 3 eq.). The mixture was stirred at RT for 18 h and quenched with a 1 M aqueous solution of HCl. The product was extracted 3 times with MTBE. The combined organic phases were dried over MgSO_4_, filtered, and concentrated under reduced pressure. A purification by flash chromatography on silica gel (12 g) with heptane/EtOAc from 8:2 to 0:10 gave compound **15k** as a yellow oil (19 mg, 0.05 mmol, 80%). ^1^H NMR (500 MHz, CDCl_3_, 25 °C): *δ* 7.97 (dd, *J* = 8.0, 1.2 Hz, 1H), 7.53 (t, *J* = 7.6 Hz, 1H), 7.31 (d, *J* = 8.0 Hz, 1H), 7.21 (t, *J* = 8.0 Hz, 1H), 6.77 (d, *J* = 10.0 Hz, 1H), 5.48 (d, *J* = 10.0 Hz, 1H), 5.09 (t, *J* = 7.0 Hz, 1H), 4.30 (t, *J* = 6.0 Hz, 2H), 2.46 (t, *J* = 6.0 Hz, 2H), 2.14 (q, *J* = 7.7 Hz, 2H), 1.89–1.70 (m, 6H), 1.62 (s, 3H), 1.55 (s, 3H), 1.47 (s, 3H) ppm. ^13^C NMR (125 MHz, CDCl_3_, 25 °C): *δ* 178.0, 160.9, 155.6, 138.4, 132.0, 130.9, 125.4, 123.7, 123.3, 121.8, 118.2, 116.2, 114.1, 105.2, 81.5, 41.8, 41.6, 33.7, 27.1, 27.0, 25.6, 22.6, 22.1, 17.6 ppm. HRMS (ESI+): calcd. for C_24_H_30_NO_4_^+^ [M+H]^+^ 396.2169, found 396.2178.

5-Benzyl-8,8,11a-trimethyl-5,8,9,10,11,11a-hexahydro-6*H*-chromeno [3,2-*c*]quinolin-6-one (**16a**). To a solution of *N*-benzyl quinolone **15a** (50 mg, 0.16 mmol, 1 eq.) in anhydrous dichloromethane (5 mL) under Ar atm. was slowly added SnCl_4_ (84 mg, 0.32 mmol, 2 eq.). After 1 h at RT, the mixture was quenched with water (0.2 mL). MgSO_4_ was added to the solution and the solids were eliminated by filtration over a Celite^®^ pad. The filtrate was concentrated under reduced pressure and purified by flash chromatography on silica gel (24 g) with heptane/EtOAc from 10:0 to 5:5 to give compound **16a** as a white solid (29 mg, 0.09 mmol, 57%). ^1^H NMR (500 MHz, CDCl_3_, 25 °C): *δ* 7.96 (dd, *J* = 7.9, 1.0 Hz, 1H), 7.36 (td, *J* = 8.0, 1.0 Hz, 1H), 7.23 (t, *J* = 8.0 Hz, 1H), 7.18 (t, *J* = 8.0 Hz, 2H), 7.13 (t, *J* = 8.0 Hz, 3H), 5.49 (bs, 2H), 4.78 (s, 1H), 2.19 (d, *J* = 13.6 Hz, 1H), 1.78 (dt, *J* = 13.6, 7.4 Hz, 1H), 1.71 (s, 1H), 1.56 (td, *J* = 13.9, 9.6 Hz, 1H), 1.48–1.45 (m, 1H), 1.40 (s, 3H), 1.29 (td, *J* = 13.9, 9.6 Hz, 1H), 1.07 (s, 3H), 0.57 (s, 3H) ppm. ^13^C NMR (125 MHz, CDCl_3_, 25 °C): *δ* 164.1, 156.4, 138.5, 136.6, 130.9, 128.8 (2C), 127.3, 126.5 (2C), 123.7, 122.0, 116.8, 114.9, 109.0, 78.6, 62.4, 52.0, 45.6, 41.3, 39.2, 33.0, 31.7, 26.7, 21.2, 17.9 ppm. HRMS (ESI+): calcd. for C_26_H_28_NO_2_^+^ [M+H]^+^ 386.2115, found 386.2125.

### 3.3. Cell Cultures

Promastigotes of *Leishmania infantum* (MHOM/FR/2008/LEM5700) were cultured in the dark at 26 °C with 5% CO_2_ in M199 complete medium containing M199 medium supplemented with adenosine (100 µM), hemin (0.5 mg/L), Hepes (40 mM) pH 7.4, and heat-inactivated fetal bovine serum (10%; HIFBS). Cultures of axenic amastigotes of *L. infantum* were adapted from [29]. Briefly, axenic amastigotes of *L. infantum* were obtained from late log promastigotes diluted at 1 × 10^6/^mL in M199 complete medium acidified at pH 5.5 and cultured at 37 °C with 5% CO_2_.

The RAW 264.7 macrophages (ATCC) were cultured at 37 °C with 5% CO_2_ in DMEM complete medium containing Dulbecco’s Modified Eagle’s Medium (DMEM, Invitrogen, Thermo-Fisher, Villebon-sur-Yvette, France) supplemented with penicillin-streptomycin (100 U/mL; Invitrogen) and heat-inactivated fetal bovine serum (10%; HIFBS).

### 3.4. In Vitro Cytotoxicity Evaluation of Compounds

Cytotoxicity was evaluated on RAW 264.7 macrophages [26]. Cells were plated in 96-well microplates at a density of 2 × 10^4^ cells per well. After an incubation of 24 h at 37 °C with 5% CO_2_, the medium was removed from each well, and 100 µL of DMEM complete medium containing two-fold serial dilutions of the compounds, from 100 µM to 0.049 µM, was added to each well. After 48 h of incubation at 37 °C with 5% CO_2_, 10 µL of resazurin (450 µM) was added to each well and further incubated in the dark for 4 h at 37 °C with 5% CO_2_. In living cells, resazurin is reduced in resorufin, and this conversion is monitored by measuring OD_570nm_ (resorufin) and OD_600nm_ (resazurin; multimode microplate reader Spark^®^, Tecan, Lyon, France). The cytotoxicity of the compounds was expressed as CC_50_ (Cytotoxic Concentration 50%: concentration inhibiting macrophage metabolic activity by 50%) and was determined by nonlinear regression in GraphPad Prism 7.0. In vitro cytotoxicity assays were performed in triplicate in three independent experiments.

### 3.5. In Vitro Antileishmanial Evaluation of Compounds on Axenic and Intramacrophage Amastigotes

The evaluations of activity on axenic amastigotes were adapted from the protocols previously described [29]. Briefly, two-fold serial dilutions of the compounds, from 100 µM to 0.049 µM, were performed in 100 µL of complete medium (see above) in 96-well microplates Axenic amastigotes were then added to each well at a density of 10^6/^mL in 200 µL final volume. After 72 h of incubation at 37 °C with 5% CO_2_, 20 µL of resazurin (450 µM) was added to each well and further incubated in the dark for 24 h at 37 °C with 5% CO_2_. Cell viability was then monitored as described above. The activity of the compounds was expressed as IC_50_, which was determined by nonlinear regression using GraphPad Prism 7.0. Miltefosine was used as the reference drug.

Concerning the evaluation on intramacrophage amastigotes, the determination of the cytotoxicity as presented above was necessary to use the highest drug concentrations to be studied on the intramacrophage amastigote model. The evaluation on the intracellular form on the parasite was performed as previously described [26]. RAW264.7 macrophages were plated in 16-well Lab-Tek chamber slides (Thermo-Fisher Villebon-sur-Yvette, France) at a density of 2 × 10^4^ cells per well and incubated for 24 h at 37 °C with 5% CO_2_. Axenic amastigotes were differentiated as described above, centrifuged at 2000× *g* for 10 min, resuspended in DMEM complete medium, and added to each well to reach a 16:1 parasite to macrophage *ratio*. After 24 h of infection at 37 °C with 5% CO_2_, extracellular parasites were removed, and DMEM complete medium (100 µL) containing two-fold serial dilutions of the compounds, from 100 µM to 0.049 µM, was added to each well. A positive control treated with 1% DMSO was added to each Lab-tek chamber slide. After 48 h of treatment, the medium was removed and cells were fixed in methanol for 1 min, stained in 10% Giemsa (Merck, Guyancourt, France) for 5 min, and further rinsed in water before observation in phase contrast at the microscope (Olympus CX31; Olympus, Rungis, France). The number of amastigotes was counted by two independent experimenters for a total of 300 macrophages per well and the *ratio* of amastigotes per macrophage was determined for each condition. This *ratio* was further compared with the one of the positive controls, considered as 100%, to determine a percentage of inhibition (% I), as follows: % I = 100 − [(*ratio* of amastigotes per macrophage in treated cells/*ratio* of amastigotes per macrophages in untreated cells) × 100]. The activity of the compounds was expressed as IC_50_, which was determined by nonlinear regression using GraphPad Prism 7.0. Miltefosine was used as the reference drug. In vitro antileishmanial evaluations were performed in triplicate in three independent experiments.

## Data Availability

All data are already provided in the manuscript.

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
