# Peer review of "Synthesis and Anti-Leishmanial Properties of Quinolones Derived from Zanthosimuline"

_molecules, 2022, doi:10.3390/molecules27227892_

Round 1

Reviewer 1 Report

1.      Please provide copies of FT-IR of the products in the manuscript and supplementary material. I could not see the spectra. In section 3.1. General Experimental Details, the authors declared that they used FT-IR for structural elucidation, but there is no FT-IR data and spectral in the manuscript and supplementary material.

2.      Please added some descriptive text about spectral data of the main products to the manuscript.

3.      For synthetic methods, the authors should add some related references.

4.      Please replace “ElectroSpray Ionization (ESI).” with “Positive ion mode electrospray ionization (ESI+)”

5.      The stereochemistry for transformation 5 to 14 and 15a to 16a are unclear. Please explain how you controlled the stereochemistry of the reactions? There is no data about stereochemistry for preparation of compound 14 in reference [12]. What is optical activity of compound 14 and 15a-15k and 16a? The author must add stereochemistry data of the reactions and the products to the manuscript (ee%, de% and optical activity). Please add the analytical data and relative descriptive text to the manuscript.

6.      Make sure that any abbreviations that are used are defined somewhere in the manuscript. EDDA,

7.      The authors done some reactions under microwave irradiations, but there is no information about microwave instrument specifications?,  power?, temperature (how it can controlled)?. Please add them to the manuscript

Author Response

Dear Reviewer,

Thank you for evaluating the manuscript entitled “Synthesis and Anti-Leishmanial Properties of Quinolones Derived from Zanthosimuline” for a submission in the journal Molecules. We did our absolute best to address your queries. You will find below the responses to all your comments. The changes in the manuscript have been made using the track change option.

  1. Please provide copies of FT-IR of the products in the manuscript and supplementary material. I could not see the spectra. In section 3.1. General Experimental Details, the authors declared that they used FT-IR for structural elucidation, but there is no FT-IR data and spectral in the manuscript and supplementary material.

No IR analysis was performed. The sentence about the use of FT-IR in the section 3.1. General Experimental Details has been added by error. The sentence has been removed.

  1. Please added some descriptive text about spectral data of the main products to the manuscript.

A paragraph has been added (line 105 to118) to discuss about this topic.

  1. For synthetic methods, the authors should add some related references.

As proposed, some references from the literature have been added.

  1. Please replace “ElectroSpray Ionization (ESI).” with “Positive ion mode electrospray ionization (ESI+)”

ESI+ has been added according to the reviewer advice.

  1. The stereochemistry for transformation 5 to 14 and 15a to 16a are unclear. Please explain how you controlled the stereochemistry of the reactions? There is no data about stereochemistry for preparation of compound 14 in reference [12]. What is optical activity of compound 14 and 15a-15k and 16a? The author must add stereochemistry data of the reactions and the products to the manuscript (ee%, de% and optical activity).Please add the analytical data and relative descriptive text to the manuscript.

Sorry for the misunderstanding. In the cited transformations, all the products are in racemic form. There is no control of the stereochemistry. All the schemes of the manuscript have been modified to remove the extra stereocenters. However, the reaction leading to molecules 10 and 12 are diastereoselective and, in these cases, the stereochemistry has been conserved to show the relative configurations. Explanations have been added in the text lines 81, 103, 130, 133.

  1. Make sure that any abbreviations that are used are defined somewhere in the manuscript. EDDA, …

The definition of the abbreviation EDDA has been added in the text of the manuscript line 83.

  1. The authors done some reactions under microwave irradiations, but there is no information about microwave instrument specifications?,  power?, temperature (how it can controlled)?. Please add them to the manuscript.

The specifications of the Microwave apparatus utilized have been added in the section 3.1. General Experimental Details.

Thank you for your kind consideration of this revised manuscript.

Reviewer 2 Report

Reviewer comments (molecules-1957414):

 I assessed the manuscript entitled “Synthesis and Anti-Leishmanial Properties of Quinolones Derived from Zanthosimuline ” submitted by Sandy Desrat et al. and found that the reported work is interesting based on Anti-Leishmanial Properties of Quinolones but it requires minor revisions.

Some of the important observations were described here: 

---There are many typographical errors were found within the manuscripts.

--The abstract of this draft does not reflect the finding of the exact work.

-- Authors should include the Anti-Leishmanial Properties of Quinolone based molecules already available into the markets. Also discuss the more examples (from last 2-5 years) based on synthesis of quinones and their developments. For e.g.  Bioorganic chemistry,2019 , 86, 137-150; RSC Advances,2022, 12 (29), 18806-18820; and RSC Advances, 2020, 10 (58), 35499-35504 etc.

-- I have seen that the mass of many molecules, calculated and found reported same as 387.2064. kindly check it and explain. It also advises verifying the many spectral data in the manuscript and in supplementary identifying a difference. Please check it and update it.

-- Cytotoxicity of compounds should be checked with normal cell lines with repetition.  

--I didn’t find the HRMS spectra and data in the supplementary file.

-- The authors should go for careful proofreading to eliminate (a) grammatical errors; (b) many typos; and also (c) remove unnecessary information and description.

In my opinion, this manuscript can’t be accepted in its current form. A revised version will be considered for future publications.

Author Response

Dear Reviewer,

Thank you for evaluating the manuscript entitled “Synthesis and Anti-Leishmanial Properties of Quinolones Derived from Zanthosimuline” for a submission in the journal Molecules. We did our absolute best to address your queries. You will find below the responses to all your comments. The changes in the manuscript have been made using the track change option.

---There are many typographical errors were found within the manuscripts.

Some errors have been corrected in the text of the manuscript.

--The abstract of this draft does not reflect the finding of the exact work.

Some of the sentences have been slightly modified in the abstract.

-- Authors should include the Anti-Leishmanial Properties of Quinolone based molecules already available into the markets. Also discuss the more examples (from last 2-5 years) based on synthesis of quinones and their developments. For e.g.  Bioorganic chemistry,2019 , 86, 137-150; RSC Advances,2022, 12 (29), 18806-18820; and RSC Advances, 2020, 10 (58), 35499-35504 etc.

Some references (5) related to quinoline derivatives with potential anti-leishmanial properties have been added including two of the references proposed.

-- I have seen that the mass of many molecules, calculated and found reported same as 387.2064. kindly check it and explain. It also advises verifying the many spectral data in the manuscript and in supplementary identifying a difference. Please check it and update it.

The spectral data have been checked and some of the figures have been changed in the description of the products.

-- Cytotoxicity of compounds should be checked with normal cell lines with repetition.  

I would appreciate to have these information but unfortunately we have no access to cytotoxicity assays on normal cells. We could not performed these experiments.

--I didn’t find the HRMS spectra and data in the supplementary file.

The HRMS spectra of all the compounds have been added in the supplementary information document.

-- The authors should go for careful proofreading to eliminate (a) grammatical errors; (b) many typos; and also (c) remove unnecessary information and description.

Some errors have been corrected in the text of the manuscript.

Thank you for your kind consideration of this revised manuscript.

Reviewer 3 Report

I greatly appreciate authors effort to develop various quinolone derivatives of Zanthosimuline and study of its biological activities.

The biological results of the reported Zanthosimuline derivatives seem to be very promising. But I have the following comments about this article.

1.Line 59: To address this question, we have developed the synthesis of natural chloro-australasine A 3 and a set of analogues to assess their anti-leishmanial potential.

The synthesis of chloro- australasine A was already reported by your group [Olivon, F.; Apel, C.; Retailleau, P.; Allard, P.M.; Wolfender, J.L.; Touboul, T.; Litaudon, M.; Desrat, S. Searching for original 520 natural products by molecular networking: detection, isolation and total synthesis of chloroaustralasines. Org. Chem. Front. 521 2018, 5, 2171-2178.].

I went through the above article, it seems all the starting materials, reagents and reaction conditions are exactly same. I felt there was no novelty in the scheme 1 or table 1 results.

Here in the current article authors synthesized various N-substituted compounds 15a-l. It seems to be analogues of Zanthosimuline not the analogues of Cloro-australasine A.

2. Most of the Table 1 was also reported in the previous article except few conditions (e.g., entries 3, 4 and 15).

Table 1, entry 7:  I wondered, the reaction conditions seem exactly same but, the yield of compound 11 was reported as 28%. Whereas in the previous paper compound 11 reported as 28%. 

Similarly, in entry 9: The yield of compound 10 was reported as 95%. Whereas in the previous paper compound 11 was reported as 92% with exactly similar reaction conditions. The results seem so mismatched with the previous results reported by your group only.

3. In scheme 2, There was no evidence of predicting the stereochemistry of compound 14. Remove the conversion of 15l. Anyway, this conversion 15k – 15I was already shown in the bottom list of analogues.

4. Melting points were not reported for the compounds which have physical state as solids. Compounds 8, 3, 11, 14.

5. Line 221: It would be nice if you mention the IUPAC name of the compound instead of compound 8 and even for other compounds. In supporting Information 13C NMR of compound 8 was missing. It would be more convenient to readers if the structure is showing up on the 1H and 13C NMR spectras.

6. 13C NMR spectra was missed for compounds 8, 14. 13C NMR spectra of compound 3 is not looking good. Please provide clean and nice spectra.

7. Optical rotations should be reported for compound 14; for compounds 15a-I.

8. References 13 and 21 are the same. Remove one of these references.

9. Based On the results shown in table 2, compound 3 seems to be more potent. This work ‘ll turn out to be very interesting if authors could generate few analogues of compound 3 from the more potent analogues of Zanthosimuline derivatives that have been reported in scheme 2.

10. This manuscript needs major revision. Remove all the reported schemes, tables etc. Please provide all the missing spectra, melting points, evidence for stereochemistry prediction and rotations for chiral compounds.

Author Response

Thank you for evaluating the manuscript entitled “Synthesis and Anti-Leishmanial Properties of Quinolones Derived from Zanthosimuline” for a submission in the journal Molecules. We did our absolute best to address your queries. You will find below the responses to all your comments. The changes in the manuscript have been made using the track change option.

1. Line 59: To address this question, we have developed the synthesis of natural chloro-australasine A and a set of analogues to assess their anti-leishmanial potential.

The synthesis of chloro- australasine A was already reported by your group [Olivon, F.; Apel, C.; Retailleau, P.; Allard, P.M.; Wolfender, J.L.; Touboul, T.; Litaudon, M.; Desrat, S. Searching for original 520 natural products by molecular networking: detection, isolation and total synthesis of chloroaustralasines. Org. Chem. Front. 521 2018, 5, 2171-2178.].

I went through the above article, it seems all the starting materials, reagents and reaction conditions are exactly same. I felt there was no novelty in the scheme 1 or table 1 results.

Here in the current article authors synthesized various N-substituted compounds 15a-l. It seems to be analogues of Zanthosimuline not the analogues of Cloro-australasine A.

To introduce the paper but also to compare with the results obtain in the previous article, we decided to report the reactions already described in scheme 1 and in table 1. I agree that we finally prepared analogues of zanthosimuline more than analogues of Chloro-australasine A. For that reason, the cited sentence line 59 has been modified.

  1. Most of the Table 1 was also reported in the previous article except few conditions (e.g., entries 3, 4 and 15).

Table 1, entry 7:  I wondered, the reaction conditions seem exactly same but, the yield of compound 11 was reported as 28%. Whereas in the previous paper compound 11 reported as 28%. 

Similarly, in entry 9: The yield of compound 10 was reported as 95%. Whereas in the previous paper compound 11 was reported as 92% with exactly similar reaction conditions. The results seem so mismatched with the previous results reported by your group only.

To compare with the results obtained in the previous article, we decided to report the conditions already described in the table. In addition, we thought it would be easier for the reader to see the structures of the evaluated compounds. For the second point, some reactions have been repeated since the previous article, giving us the opportunity to improve the yield of obtention of some of the compounds. I confirm that compound 10 was obtained in 95% yield in the last reaction we made.

  1. In scheme 2, There was no evidence of predicting the stereochemistry of compound 14. Remove the conversion of 15l. Anyway, this conversion 15k – 15I was already shown in the bottom list of analogues.

Sorry for the mistake. In the reactions described, all the products are in racemic form. There is no control of the stereochemistry. All the schemes of the manuscript have been modified to remove the extra stereocenters. Some explanations have been added in the text lines 81, 103, 130, 133. The transformation of 15l to 15k was removed from the scheme.

  1. Melting points were not reported for the compounds which have physical state as solids. Compounds 8, 3, 11, 14.

We consider reporting melting points only when products are crystalline, not when they are amorphous. We have no longer the possibility to measure it anyway.

  1. Line 221: It would be nice if you mention the IUPAC name of the compound instead of compound 8and even for other compounds. In supporting Information 13C NMR of compound was missing. It would be more convenient to readers if the structure is showing up on the 1H and 13C NMR spectra.

The IUPAC name of all the compounds have been added in the description of the compounds. The structure of the compounds have been added on the spectra in the supplementary material.

  1. 13C NMR spectra was missed for compounds 8, 14. 13C NMR spectra of compound is not looking good.Please provide clean and nice spectra.

13C NMR spectra of compounds 8 and 14 were not acquired as the compounds were already described in the literature. Having identical 1H NMR and HRMS spectra to what is described in the literature, we considered the structure OK. In addition, as mentioned in the text, compound 8 was not stable and degradation was observed in a few days if not engaged in the next reaction. The baseline of the 13C NMR spectrum of compound 3 was corrected.

  1. Optical rotations should be reported for compound 14; for compounds 15a-I.

The compounds are racemic. Schemes have been changed. No optical rotation was observed.

  1. References 13 and 21 are the same. Remove one of these references.

Ref 21 has been removed.

  1. Based on the results shown in table 2, compound seems to be more potent. This work ‘ll turn out to be very interesting if authors could generate few analogues of compound 3from the more potent analogues of Zanthosimuline derivatives that have been reported in scheme 2.

What is important is to have a compound active against the parasite in the cell (intramacrophagic amastigote). As shown in table 2, compound 3 is not active against the parasite in the host cell, contrary to compound 11. That’s why we decided to design analogues of compound 11. For that, we prepared analogues of compounds 7 (15a-l) and we tried to convert them into cyclized compound 16a-l.

  1. This manuscript needs major revision. Remove all the reported schemes, tables etc. Please provide all the missing spectra, melting points, evidence for stereochemistry prediction and rotations for chiral compounds.

Thank you very much for revising this manuscript. We did our absolute best to address your queries as detailed above.

Round 2

Reviewer 1 Report

Dear Editor
The paper was revised according to the reviewer’ comments.
In its current state it is ready for publication in your journal.
Best regards

Author Response

Dear Reviewer,

Thank you for your kind consideration of this revised manuscript entitled “Synthesis and Anti-Leishmanial Properties of Quinolones Derived from Zanthosimuline” for a submission in the journal Molecules. 

Best regards

Dr Sandy Desrat

Reviewer 3 Report

Dear Authors,

I appreciate that the manuscript was modified according to the reviewer's comments.

1. But there is still lack of the scientific soundness due to the presentation of reported methods and reaction protocols and spectras etc. There is no novelty in this work.

2. Though authors have tried few new conditions in the entries 3,4,6,10,11 in table 1. None of them improved the yields or selectivity of the compounds 8, 9, 10, 11 except entry 15.

3.The conversion from compound 15 to compound 16 was also failed in many cases except with benzyl derivatives. Authors haven't stabilized the conditions.

4. The biological results were interesting. Authors can publish these results in any other suitable journal. 

5. Due to the above concerns, I am rejecting this manuscript to publish in Molecules.

Author Response

Dear Reviewer,

Thank you for your kind consideration of this revised manuscript. I understand your point of view. However, I consider that the first schemes of the manuscript introduce our work for a better understanding. In addition, most of molecules described are new, and their leishmanicidal activities have never been described. So I don't consider there is no novelty in this work. 

Thank you again for your comments

Best regards

Dr Sandy Desrat